# Systematic Analysis of Long Non-Coding RNAs in Inflammasome Activation in Monocytes/Macrophages

**DOI:** 10.3390/ncrna9050050

**Published:** 2023-08-25

**Authors:** Na Qian, Rebecca Distefano, Mirolyuba Ilieva, Jens Hedelund Madsen, Sarah Rennie, Shizuka Uchida

**Affiliations:** 1Department of Biology, University of Copenhagen, DK-2200 Copenhagen N, Denmark; rqf897@alumni.ku.dk (N.Q.); fwh492@alumni.ku.dk (R.D.); sarah.rennie@bio.ku.dk (S.R.); 2Center for RNA Medicine, Department of Clinical Medicine, Aalborg University, DK-2450 Copenhagen SV, Denmark; mirolyubasi@dcm.aau.dk (M.I.); jenshm@dcm.aau.dk (J.H.M.)

**Keywords:** inflammasome, lncRNA, macrophages, RNA-seq

## Abstract

The NLRP3 inflammasome plays a pivotal role in regulating inflammation and immune responses. Its activation can lead to an inflammatory response and pyroptotic cell death. This is beneficial in the case of infections, but excessive activation can lead to chronic inflammation and tissue damage. Moreover, while most of the mammalian genome is transcribed as RNAs, only a small fraction codes for proteins. Among non-protein-coding RNAs, long non-coding RNAs (lncRNAs) have been shown to play key roles in regulating gene expression and cellular processes. They interact with DNA, RNAs, and proteins, and their dysregulation can provide insights into disease mechanisms, including NLRP3 inflammasome activation. Here, we systematically analyzed previously published RNA sequencing (RNA-seq) data of NLRP3 inflammasome activation in monocytes/macrophages to uncover inflammasome-regulated lncRNA genes. To uncover the functional importance of inflammasome-regulated lncRNA genes, one inflammasome-regulated lncRNA, *ENSG00000273124*, was knocked down in an in vitro model of macrophage polarization. The results indicate that silencing of *ENSG00000273124* resulted in the up-regulation tumor necrosis factor (*TNF*), suggesting that this lncRNA might be involved in pro-inflammatory response in macrophages. To make our analyzed data more accessible, we developed the web database InflammasomeDB.

## 1. Introduction

Inflammasomes are complex protein assemblies found in macrophages, dendritic cells, and neutrophils [1,2,3,4,5]. They control the activation of cysteine protease caspase-1. This enzyme then regulates the maturation of pro-interleukin (IL)-1β and IL-18, leading to a process called pyroptosis. The NLRP3 inflammasome, comprising NLRP3, the adaptor molecule caspase-recruitment domain (ASC), and the cysteine protease caspase-1, plays a crucial role in regulating inflammation and immune responses. In the context of infections or injuries, this response is crucial for host defense and tissue regeneration. In conditions of sterile inflammation, overactivation, or extended activation of the NLRP3 inflammasome, leading to increased production of IL-1β and IL-18, they can result in persistent inflammation and harm to tissues [2].

Although most of the mammalian genome is transcribed as RNA, the majority of these transcripts are non-protein-coding RNAs (ncRNAs), while only a small fraction corresponds to exons of protein-coding genes [6]. Within the realm of ncRNAs, long non-coding RNAs (lncRNAs) have emerged as significant regulators of gene expression and cellular processes. Unlike protein-coding RNAs, lncRNAs lack the ability to generate proteins but exert crucial roles in diverse biological processes through interactions with DNA, RNA, and proteins [7,8]. Additionally, due to their cell-type specific expression patterns and frequent dysregulation in the disease, lncRNAs present unprecedented opportunities for elucidating disease mechanisms [9,10], including NLRP3 inflammasome activation [11,12,13].

Here, we revisited the previously published RNA sequencing (RNA-seq) data of NLRP3 inflammasome activation by focusing especially on those data derived from monocytes/macrophages. By performing secondary analyses on previously generated RNA-seq data, which were analyzed only for protein coding but not lncRNA genes, we uncovered distinctive expression profiles of lncRNA genes. These profiles were found to be consistent across the different analyzed datasets. To demonstrate the validity of our data analysis, we activated the NLRP3 inflammasome in the human leukemia monocytic cell line, THP-1, to record the expression changes of inflammasome-regulated lncRNA genes. Since many lncRNAs have not been functionally studied yet, we performed loss-of-function experiments on the inflammasome-regulated lncRNA, *ENSG00000273124*. Finally, to facilitate further research of inflammasome-regulated lncRNAs, we built a web application, InflammasomeDB (https://qianna0321.shinyapps.io/inflammasomedb/, accessed on 5 August 2023), for expression profiles of protein-coding and lncRNA genes related to the NLRP3 inflammasome activation.

## 2. Results

### 2.1. CXCL4 and TLR8 Co-Stimulation Induces the Expression of lncRNA Genes

Monocytes and macrophages are types of white blood cells (leukocytes) involved in innate immunity [14,15]. Monocytes are large, circulating cells that act as precursors to macrophages. In tissues, monocytes change into macrophages, which perform phagocytosis to eliminate pathogens and cellular debris. They also release cytokines, regulate immune responses, and contribute to tissue repair and homeostasis. In multiple inflammatory and fibrotic diseases, the chemokine CXCL4 (official gene name, platelet factor 4 (*PF4*)) promotes inflammation by amplifying the Toll-like receptor (TLR) signaling (especially Toll-like receptor 8 (TLR8)) of innate and adaptive immune cells via the up-regulation of inflammatory gene transcription [16,17]. To elucidate gene regulation by CXCL4 and TLR8, Yang et al. performed RNA-seq using primary human blood monocytes treated with CXCL4 and the TLR8 single-stranded RNA (ssRNA) ligand, ORN8L, to activate TLR8 (Gene Expression Omnibus (GEO) accession number, GSE181889) [17]. The original study uncovered that CXCL4 and TLR8 co-stimulation of monocytes activates a TANK binding kinase 1 (*TBK1*)-interferon regulatory factor 5 (*IRF5*) axis to drive inflammatory gene expression to activate the NLRP3 inflammasome. Since the original study analyzed the expression profiles of protein-coding genes but not of lncRNAs, we further dissected the original data to uncover the expression changes of lncRNA genes.

By applying the threshold values of twofold and false discovery rate (FDR)-adjusted *p*-values < 0.05, there are many differentially expressed genes identified upon treatment with ORN8L (but not with CXCL4) as reported in the original study [17] (Figure 1A), including those of lncRNA genes (Figure 1B). The number of differentially expressed genes is much higher when the cells were treated with both CXCL4 and ORN8L as reported previously. Because Gene Ontology (GO) and pathway analyses of differentially expressed genes were already performed in the original study, which identified the enrichments of immune responses and cytokine signaling [17], we focused solely on lncRNA genes instead of protein-coding genes. Although there are significant numbers of both up- and down-regulated lncRNA genes shared between the two comparisons (Figure 1C; Appendix A), the treatment with both CXCL4 and ORN8L showed much higher numbers of distinctive lncRNA genes as was the case for protein-coding genes in the original study, suggesting that the expression of lncRNA genes can be altered by CXCL4 and TLR8 co-stimulation leading to inflammation in monocytes to activate the NLRP3 inflammasome.

### 2.2. Distinctive Sets of lncRNA Genes Were Induced by LPS and MSU Treatments in Macrophages

Macrophages are highly plastic cells as they can switch from a basal status, where they carry out tissue-specific homeostatic functions, to an inflammatory state, where they possess the capacity to eliminate invading pathogens and dead cells [18,19]. To understand the gene expression changes associated with macrophage plasticity, Cobo et al. induced murine and human macrophages with lipopolysaccharide (LPS, an outer membrane component of Gram-negative bacteria [20]) to activate the NLRP3 inflammasome or monosodium urate (MSU) crystals to induce inflammation [21]. Through RNA-seq assay (GEO accession number, GSE191054), the authors uncovered that compared to LPS-treated cells, MSU alone induced a metabolic-inflammatory transcriptional program, including lipid and amino acid metabolism, glycolysis, and solute carrier (SLC) transporters. However, they did not analyze for lncRNAs; therefore, we re-analyzed the human RNA-seq data to derive differentially expressed lncRNA genes by applying the threshold values of twofold and FDR < 0.05.

As shown in the original study [21], LPS treatment resulted in more differentially expressed genes than MSU treatment compared to the phosphate-buffered saline (PBS)-control samples (Figure 2A). Yet, there are significant differences between LPS and MSU treatments, even for the numbers of differentially expressed lncRNA genes (Figure 2B). When differentially expressed lncRNA genes were compared between LPS and MSU treatments and also compared to the control samples, some up- and down-regulated lncRNA genes were demonstrated, although they are only small portions compared to LPS or MSU alone (Figure 2C; Appendix A), which reflect the plasticity of macrophages based on the stimuli. Although the functions of most shared lncRNA genes are not known, some lncRNAs have been implicated to be involved in tumorigenesis in various cancers (e.g., long intergenic non-protein coding RNA, p53 induced transcript (LINC-PINT) [22] and cytoskeleton regulator RNA (CYTOR) [23]). However, to the best of our knowledge, nothing is known about their involvement in macrophage plasticity leading to the NLRP3 inflammasome activation, which calls for further investigations.

### 2.3. Expression and Functional Analysis of Inflammasome-Regulated lncRNAs

It is evident from the above data analyses that there are many up-regulated genes in both monocytes and macrophages after NLRP3 inflammasome activation; some of which are shared among different conditions (Figure 3A; Appendix A). To date, only a few lncRNAs have been studied in relation to the NLRP3 inflammasome activation in inflammatory cells, especially in macrophages [24,25,26,27,28,29,30,31,32,33,34,35]. To address this point, we selected both protein-coding and lncRNA genes whose expressions were up-regulated in the above-analyzed RNA-seq data (Appendix A). The protein-coding genes are CASP8 and FADD-like apoptosis regulator (*CFLAR*), LIF interleukin 6 family cytokine (*LIF*), the regulator of G protein signaling 1 (*RGS1*), and radical S-adenosyl methionine domain containing 2 (*RSAD2*), while lncRNA genes are long intergenic non-protein-coding RNA 1181 (*LINC01181*), long intergenic non-protein-coding RNA 1215 (*LINC01215*), novel transcript, antisense to BCAT1 (*ENSG00000255921*), novel transcript *ENSG00000273124*, and novel transcript, sense intronic to GBP2 (*ENSG00000289582*). First, we confirmed their expressions in the human leukemia monocytic cell line, THP-1, treated with phorbol 12-myristate 13-acetate (PMA) to prime these cells into macrophage-like cells (designated as M (-)). Next, to activate the NLRP3 inflammasome, the M (-) cells were induced with lipopolysaccharide (LPS) (designated as M (LPS)) [36]. As shown in Figure 3B, all selected commonly up-regulated protein-coding and three out of five lncRNA genes showed statistically significant up-regulation in M (LPS) compared to M (-) cells as in the case of inflammasome markers (caspase 1 (*CASP1*), interleukin 1 beta (*IL1B*), and *NLRP3*).

### 2.4. Subcellular Localization of lncRNAs May Indicate Their Functionalities

Compared to mature messenger RNAs (mRNAs), which are found in the cytosol [37], lncRNAs can be found in both the nuclear and cytosol of a cell. This distinctive subcellular localization may indicate the functions of lncRNAs as nuclear lncRNAs are known to regulate epigenetic and transcriptional activities, while cytosolic lncRNAs function as miRNA or RNA-binding protein (RBP) sponges [38,39,40]. To understand the subcellular localization of lncRNAs and their interactions with proteins, Aznaourova et al. performed RNA-seq experiments in LPS-treated human primary macrophages by separating nuclear and cytosol fractions (GEO accession number, GSE101409) [41]. By re-analyzing this valuable dataset with the same analysis pipeline as the above two studies, we further annotated the inflammasome-regulated lncRNA genes based on their subcellular localization (Appendix A). Of 324 shared up-regulated lncRNA genes (in at least two conditions compared), 17 lncRNA genes are enriched in the nuclear fraction, while 4 lncRNA genes are enriched in the cytosol with the threshold of twofold and FDR < 0.05. Upon LPS treatment, 42 lncRNA genes are enriched in the nuclear fraction, while 22 lncRNA genes are enriched in the cytosol. Of note, all of the previously selected inflammasome-regulated lncRNA genes are expressed both in the nuclear and cytosolic fractions of macrophages. Although more lncRNA genes are expressed upon LPS treatment, most shared up-regulated lncRNA genes are expressed in both nuclear and cytosolic fractions of macrophages regardless of LPS treatment.

### 2.5. Polarization of Macrophages to Pro- and Anti-Inflammatory Macrophages

Macrophages can be polarized into the pro-inflammatory subtype, M1, and the anti-inflammatory subtype, M2 [42]. The cascade of differentiation and polarizations are regulated by the coordinated actions of cytokines. In particular, the NLRP3 inflammasome mediates the polarization of naïve (M0) macrophages towards M1 but not M2 [43]. To understand the cell signaling and metabolism during macrophage polarization, He et al. performed transcriptomic and proteomic assays, including RNA-seq analysis of human peripheral blood mononuclear cells (PBMC) and THP-1 cells treated with interferon gamma (IFNγ)/LPS and IL-4 to polarize the cells towards M1 and M2 macrophages, respectively (GEO accession numbers, GSE154345 and GSE154346) [44]. As a major focus of this study was on proteins, we performed a secondary analysis of RNA-seq data of PBMC and THP-1 cells (Figure 4A,B).

When the expression profiles of 324 shared up-regulated lncRNA genes were screened, 61 were highly up-regulated in M1 compared to M2 macrophages in both PBMC and THP-1 cells (Figure 4C; Appendix A). For those selected protein-coding and lncRNA genes shown in Figure 3A, a majority of them are expressed much higher in M1 compared to M2 macrophages (Figure 4D), supporting the previous data analyses for the NLRP3 inflammasome activation leading to the polarization of M0 macrophages to M1 macrophages.

### 2.6. Loss-of-Function Experiments in Pro-Inflammatory Macrophages

As shown in the previous subsection, our inflammasome-regulated lncRNA genes were highly up-regulated during the polarization of naïve (M0) macrophages to pro-inflammatory M1 macrophages. To further confirm these findings, the expression changes of inflammasome-regulated protein-coding and lncRNA genes were recorded in an in vitro model of pro-inflammatory macrophage activation by stimulating THP-1 cells to macrophage-like cells (M (-); similar to M0 macrophages) and then to pro-inflammatory macrophage-like cells (M (IFN-γ/LPS); similar to M1 macrophages), as described in the Materials and Methods Section. As shown in Figure 5A, all genes (except the protein-coding genes, *LIF*, and the lncRNA gene, *LINC01215)* were found to be highly up-regulated in pro-inflammatory M (IFN-γ/LPS) cells compared to M (-) cells.

To assess the functional importance of inflammasome-regulated lncRNA genes, we silenced the inflammasome-regulated lncRNA gene, *ENSG00000273124*. Compared to the control samples (transfected with siRNA against random sequences; siScr), the expression of *ENSG00000273124* were significantly down-regulated by siRNA (Figure 5B). When the markers of inflammation were quantified, silencing of *ENSG00000273124* resulted in up-regulation of inflammatory marker genes, tumor necrosis factor (TNF), and major histocompatibility complex, class II, DR alpha (HLA-DRA) (Figure 5C), suggesting that this inflammasome-regulated lncRNA gene might be involved in the polarization of macrophage-like cells to pro-inflammatory macrophage-like cells. Further research is necessary to uncover the mechanism of action of *ENSG00000273124*.

### 2.7. The Web Database, InflammasomeDB, for Expression Analysis of Inflammasome-Regulated Protein-Coding and lncRNA Genes

We developed the InflammasomeDB web database to share the results from this study and support future research on inflammasome-regulated genes (Figure 6A). This freely accessible web database is available online at https://qianna0321.shinyapps.io/inflammasomedb/ (accessed on 3 August 2023). It includes expression profiles for all protein-coding and lncRNA genes, offering a useful platform for condition-specific expression analysis, such as lncRNA genes induced by co-stimulation with CXCL4 and TLR8 in monocytes.

The database is structured into four primary pages: Explore, Download, lncRNAs, and Documentation. In the Explore section, users can effortlessly examine expression changes of protein-coding and lncRNA genes across various conditions shown in the previous subsections. Results are neatly displayed in a “Result Table” on the left side of the page (Figure 6B), and the users can customize this dynamic page using fold change (log2FC) and FDR values to control the number of differentially expressed genes (DEGs) identified. Further investigation into the DEGs is possible through a volcano plot and heat map (Figure 6B), with the volcano plot being directly connected to the “Result Table”. The users can highlight a specific gene from the table on the plot to quickly identify the gene of interest on the volcano plot. In addition, DEGs can be examined for enriched gene ontology (GO) terms (Figure 6C) and Kyoto Encyclopedia of Genes and Genomes (KEGG) pathways (Figure 6D), with clear distinctions between up- and down-regulated genes. These functions allow us to further examine the biological processes and signaling pathways, respectively, in each condition. Furthermore, the “Comparisons Intersection” window on the Explore page offers a visual representation of shared DEGs across different comparisons using a Venn diagram, with a text output listing these shared DEGs (Figure 6D). These shared DEGs can be further explored using their Ensembl Gene IDs.

The lncRNAs section provides further insight into the lncRNA genes identified in this study. The lncRNA Table details the lncRNA genes selected from each study and comparison using threshold values of a twofold change and FDR < 0.05. It contains information on conservation status, miRNA targets, genome-wide association studies (GWAS)-associated terms (the information was obtained from LncBook 2.0, a curated resource of human lncRNAs [45]), the closest protein-coding gene, and the top gene correlated with the selected lncRNA gene (Figure 6E). For users’ convenience, the four datasets used in this study are downloadable from the Download section. Comprehensive instructions for database usage, information on the datasets used, and feedback guidelines are available in the Documentation section.

## 3. Discussion

The key findings of this study are: (1) The NLRP3 inflammasome activation in monocytes/macrophages resulted in differential expression of several hundreds of lncRNA genes; (2) Despite different stimuli for the NLRP3 inflammasome activation in monocytes/macrophages, there are commonly up-regulated protein-coding and lncRNA genes, whose expression changes can be validated experimentally using THP-1 cells induced with the NLRP3 inflammasome inducer, LPS; (3) Because the NLRP3 inflammasome activation leads to the polarization of naïve macrophages to pro-inflammatory macrophages, 61 out of 324 shared inflammatory-regulated lncRNA genes showed significant up-regulation in M1 than M2 macrophages; (4) The silencing of the inflammasome-regulated lncRNA gene, *ENSG00000273124,* resulted in the up-regulation of inflammation marker genes during the polarization of macrophage-like cells into pro-inflammatory macrophage-like cells, suggesting that *ENSG00000273124* is involved in inflammation of macrophages. According to the latest annotation provided by the Ensembl database [46], *ENSG00000273124* is a novel transcript located at Chromosome 10: 90,997,480–90,997,713 on the reverse strand with only one transcript (ENST00000607979.1), which is 234 nucleotides long. As nothing is known about this lncRNA, further functional and mechanistic studies are required to uncover the exact mechanism in which *ENSG00000273124* contributes to the polarization of macrophages to pro-inflammatory macrophages initiated by the NLRP3 inflammasome activation.

To address the limitations of this study, it is important to acknowledge that the RNA-seq data analyzed was sourced from diverse laboratories utilizing varying protocols. We have tried to minimize inconsistencies arising from the various sources by individually assessing each study and comparing only the differentially expressed genes via a parallel analysis pipeline. Additionally, these RNA-seq data rely on poly-A sequencing, meaning RNAs lacking poly-A tails were omitted from the analysis. Thus, it is likely that the count of inflammasome-regulated lncRNA genes was underestimated since over half of lncRNA genes lack poly-A tails [34].

To facilitate further research in inflammasome-regulated lncRNA genes, we have developed a useful web database, InflammasomeDB. This bioinformatic tool allows researchers to explore expression profiles of both protein-coding and lncRNA genes in monocytes/macrophages triggered by the external stimuli to activate the NLRP3 inflammasome. InflammasomeDB also provides in-depth information on identified differentially expressed lncRNA genes, which may be informative for future experiments. We plan to ensure InflammasomeDB’s relevance by routinely updating it, encouraging user feedback and integrating further datasets, enabling investigations of a wider set of lncRNA candidates. Due to its user-friendliness and rich contents, we expect InflammasomeDB to be an invaluable resource for studying the role of lncRNA genes in NLRP3 inflammasome activation in monocytes/macrophages.

## 4. Materials and Methods

### 4.1. RNA-Seq Data Analysis and Visualization

The raw RNA-seq data were obtained from the Sequence Read Archive (SRA) database using SRA Toolkit (https://trace.ncbi.nlm.nih.gov/Traces/sra/sra.cgi?view=software, accessed on 5 May 2023). Following the conversion of the .sra files to FASTQ format, the data underwent preprocessing using fastp [47] (versions 0.21.0 and 0.23.2) with default settings. This involved quality control, adapter trimming, quality-based filtering, and read pruning. For mapping the trimmed sequencing reads to the reference genome (GRCh38.109), STAR [48] (versions 2.5.0a and 2.7.9a) was employed. The R package edgeR [49] (version 3.30.3) was used to calculate counts per million (CPM) values and identify differentially expressed genes. False discovery rate (FDR)-adjusted *p*-values obtained through the Benjamini–Hochberg method were utilized for subsequent analysis. The commands and programs utilized in this study can be found in the GitHub repository (https://github.com/heartlncrna/Analysis_of_Inflammasome_Study, accessed on 5 August 2023).

Volcano plots were generated using the R-package ggplot2 [50]. Overlapping genes were determined using http://bioinformatics.psb.ugent.be/webtools/Venn/ (accessed on 18 July 2023). To identify enriched gene ontology (GO) terms and Kyoto Encyclopedia of Genes and Genomes (KEGG) pathways, the Database for Annotation, Visualization, and Integrated Discovery (DAVID) (version v2022q3) [51,52] was employed. Enrichment analysis was performed for GOTERM_BP_DIRECT (GO biological process) and KEGG_PATHWAY categories, utilizing *p*-values computed by DAVID through Fisher’s Exact test. The top 10 enriched GO terms and KEGG pathways were selected based on the -logarithm (base 2) of *p*-values.

### 4.2. Cell Culture

The THP-1 human leukemia monocytic cell line (LGC Standards GmbH, Wesel, Germany, catalog #ATCC-TIB-202; Lot #70043382) was grown in an RPMI 1640 medium (Thermo Fisher Scientific, Roskilde, Denmark, catalog #21875091) supplemented with 10% fetal bovine serum (Merck Life Science, Espoo, Finland, catalog #F4135), 1% L-Glutamine solution (Merck Life Science, catalog #G7513), and 1% Penicillin-Streptomycin (Merck Life Science, catalog #P4333). The cells were maintained at a temperature of 37 °C with a 5% CO_2_ atmosphere.

To differentiate the THP-1 cells into macrophage-like cells (designated as M (-)), they were exposed to a growth medium containing 100 nM of phorbol 12-myristate 13-acetate (PMA; Sigma-Aldrich, St. Louis, MO, USA, #P8139-1MG) for a duration of 3 days. Then, to activate NLRP3 inflammasome, the M (-) cells were incubated with a growth medium supplemented with 250 ng/mL of eBioscience Lipopolysaccharide (LPS) Solution (500× concentration; 2.5 mg/mL) (Thermo Fisher Scientific, #00-4976-93). The derived cells were designated as M (LPS).

To derive activated macrophage-like cells (M (IFN-γ/LPS)), M (-) cells were incubated with the growth medium supplemented with 20 ng/mL of recombinant human interferon-γ (IFN-γ; Cell Signaling, Danvers, MA, USA, #80385) and 250 ng/mL of eBioscience Lipopolysaccharide (LPS) Solution (500×; 2.5 mg/mL) (Thermo Fisher Scientific, #00-4976-93) for 48 h.

To silence the expression of the target lncRNA gene, *ENSG00000273124*, siRNA was designed with RNAXS (http://rna.tbi.univie.ac.at/cgi-bin/RNAxs/RNAxs.cgi, accessed on 19 June 2023) and synthesized at Merck Life Science as MISSION siRNA: target sequence, CAGTCCTAGCCAATGAATA; sense-CAGUCCUAGCCAAUGAAUA[dT][dT]; and antisense-UAUUCAUUGGCUAGGACUG[dT][dT]. As a control, Mission Negative control SIC-002 (with a confidential sequence) obtained from Merck Life Science was used. To initiate gene silencing, siRNA transfection was performed on the M (LPS) cells 30 min after the activation of NLRP3 inflammasome. The final concentration of siRNA used was 50 nM using Lipofectamine™ RNAiMAX Transfection Reagent (Thermo Fisher Scientific, #13778150) as the transfection agent, following the manufacturer’s protocol. After siRNA transfection, the samples were collected 48 h later for total RNA isolation and further analysis.

### 4.3. Isolation of Total RNA and RT-PCR

The total RNA was isolated and purified using the TRIzol Reagent (Thermo Fisher Scientific, Roskilde, Denmark, #15596018) following the manufacturer’s instructions. For the synthesis of the first-strand complementary DNA (cDNA), the SuperScript IV VILO Master Mix with ezDNase Enzyme (Thermo Fisher Scientific, #11766500) was used to digest the genomic DNA and reverse transcribe the total RNA. Following the reverse transcription reaction, the cDNA samples were diluted with DNase/RNase-free water to a concentration of 1 ng/μL.

For quantitative reverse transcription polymerase chain reaction (qRT-PCR), each reaction utilized 1 ng of cDNA template. The qRT-PCR reaction was carried out using the PowerUp SYBR Green Master Mix (Thermo Fisher Scientific, #A25777) on the QuantStudio 6 Flex Real-Time PCR System (Thermo Fisher Scientific). The annealing temperature during amplification was set to 60 °C. Relative fold expression was calculated using the 2^−ΔΔCt^ method, with ribosomal protein lateral stalk subunit P0 (*RPLP0*) serving as the internal control.

The primer pairs for qRT-PCR were designed using Primer3 (http://bioinfo.ut.ee/primer3-0.4.0/, accessed on 10 June 2023) [53]. Before extensive testing, the primer pairs were validated in silico using the UCSC In-Silico PCR tool (https://genome.ucsc.edu/cgi-bin/hgPcr, accessed on 10 June 2023). Additionally, conventional RT-PCR reactions were performed to verify the primer specificity, followed by gel electrophoresis to confirm the presence of a single band of the expected size for each primer pair. Please refer to Appendix A for the primer sequences.

### 4.4. Inflammasome Web Database

The InflammasomeDB web database was developed utilizing the R package Shiny [54], similar to our previously built databases, DoxoDB [55] and T2DB [56]. InflammasomeDB has four main sections: Explore, lncRNAs, Download, and Documentation. The users can navigate these sections using the top navigation bar. The primary interactive Result table, generated with the R package DT (https://github.com/rstudio/DT, accessed on 31 July 2023), is housed on the left side of the Explore page. This table illustrates the outcomes of RNA-seq data analysis and offers additional exploration opportunities via five tabs on the page’s right side. These include the Volcano plot tab (developed with the R package ggplot2 [50]), the Heatmap tab (utilizing the pheatmap function from ComplexHeatmaps [57]), the GO analysis tab (via gprofiler2 [58]), the Pathway analysis section (using the enrichKEGG function from clusterProfiler [59,60] and enrichplot [61] for visualization), and the Comparisons Intersection tab (displayed through a Venn diagram using VennDiagram [62]).

The lncRNAs section presents annotations for the differentially expressed lncRNA genes (with |log2FC| > 1 and FDR < 0.05) identified in each study and comparison. In this section, the users can access and download the interactive lncRNA Table (created with DT) in .tsv format, along with any information shown in the text boxes on the right side of the page.

From the Download section, the users can obtain the study’s datasets in either .csv or .tsv format, previewed in a DT-rendered table. The Documentation section provides instructions on using the web application and additional information on the datasets.

The complete code utilized to establish InflammasomeDB is publicly accessible in the GitHub repository at https://github.com/ccc0321/InflammasomeDB/ (visited on 5 August 2023). The users can access InflammasomeDB freely and without the need for a password at (https://qianna0321.shinyapps.io/inflammasomedb/ (visited on 05 August 2023).

## Figures and Tables

**Figure 1 ncrna-09-00050-f001:**
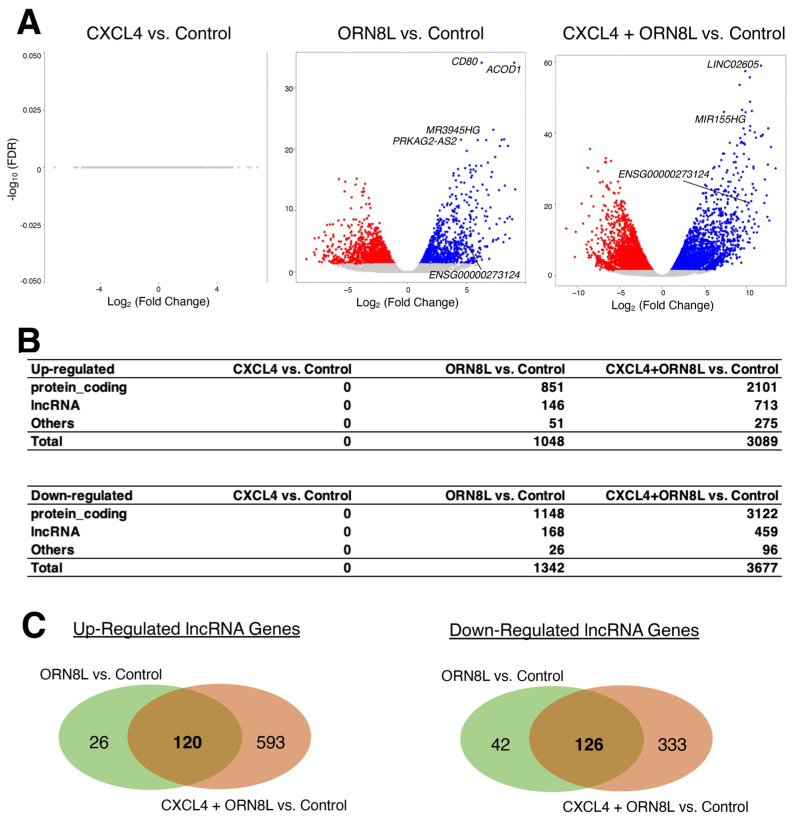
Differentially expressed genes upon treatment with CXCL4, ORN8L, or both. (**A**) Volcano plots compared to the control, resting cells. n = 3 per condition. (**B**) Numbers of up- and down-regulated genes for protein-coding, lncRNA, and other genes. Other genes include microRNAs (miRNAs), pseudogenes, ribosomal RNAs (rRNAs), and others based on the categories provided by the Ensembl database. (**C**) Venn diagrams of up- and down-regulated lncRNA genes comparing ORN8L and CXCL4 + ORN8L compared to the control.

**Figure 2 ncrna-09-00050-f002:**
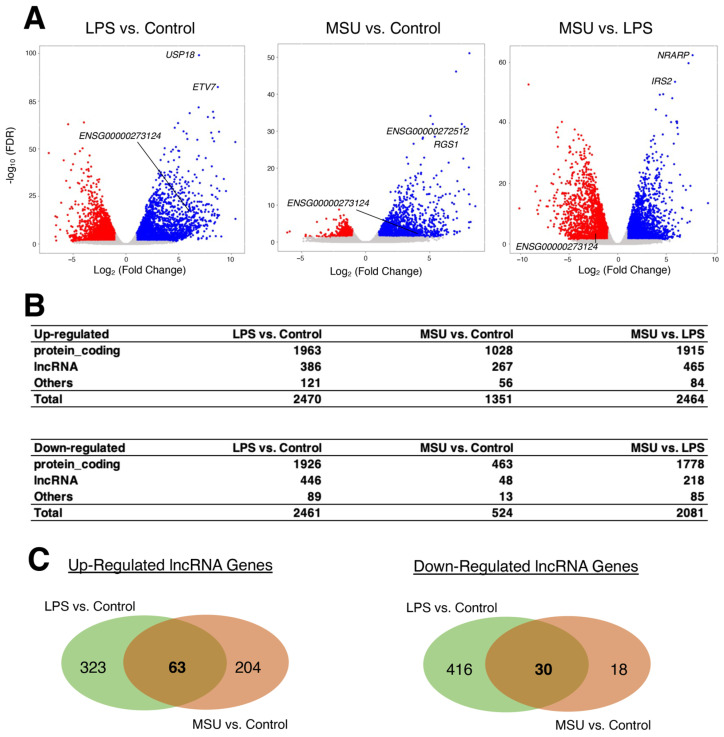
Differentially expressed genes upon treatment with LPS or MSU. (**A**) Volcano plots of differentially expressed genes in treated cells compared to the PBS control samples. The numbers of samples are 5 for control, 7 for LPS, and 5 for MSU. (**B**) Numbers of up- and down-regulated genes for protein-coding, lncRNA, and other genes. (**C**) Venn diagrams of up- and down-regulated lncRNA genes comparing LPS- and MSU-treated samples compared to the control samples.

**Figure 3 ncrna-09-00050-f003:**
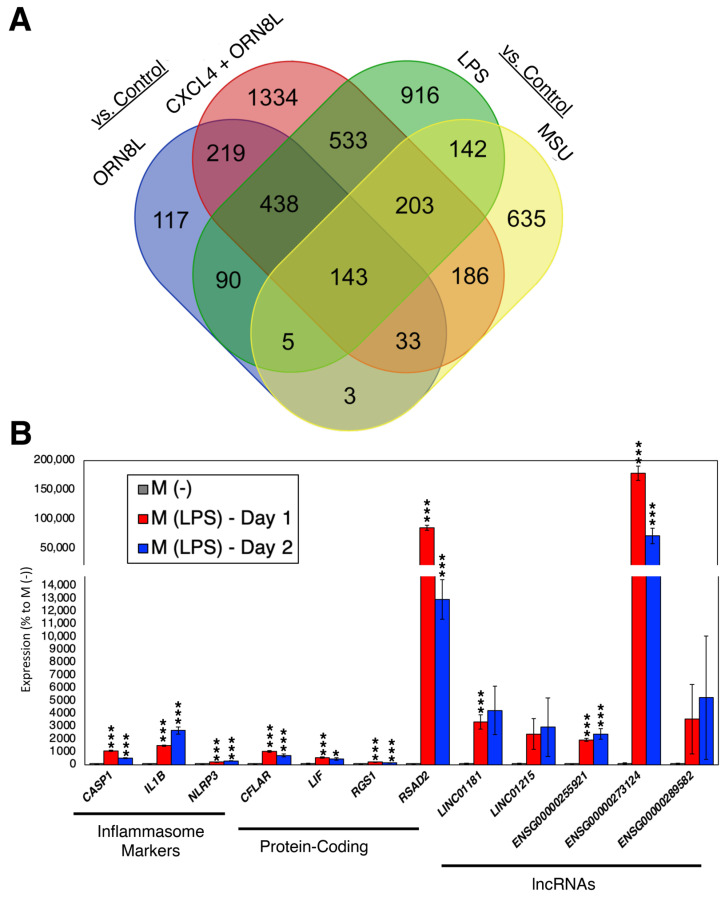
Expression analysis of NLRP3 inflammasome-regulated genes. (**A**) Venn diagram showing differentially expressed genes identified in 4 comparisons from 2 datasets analyzed in the previous subsections. (**B**) Expression profiles of LPS-regulated genes. n = 6 biological replicates. * (*p* < 0.05) and *** (*p* < 0.005).

**Figure 4 ncrna-09-00050-f004:**
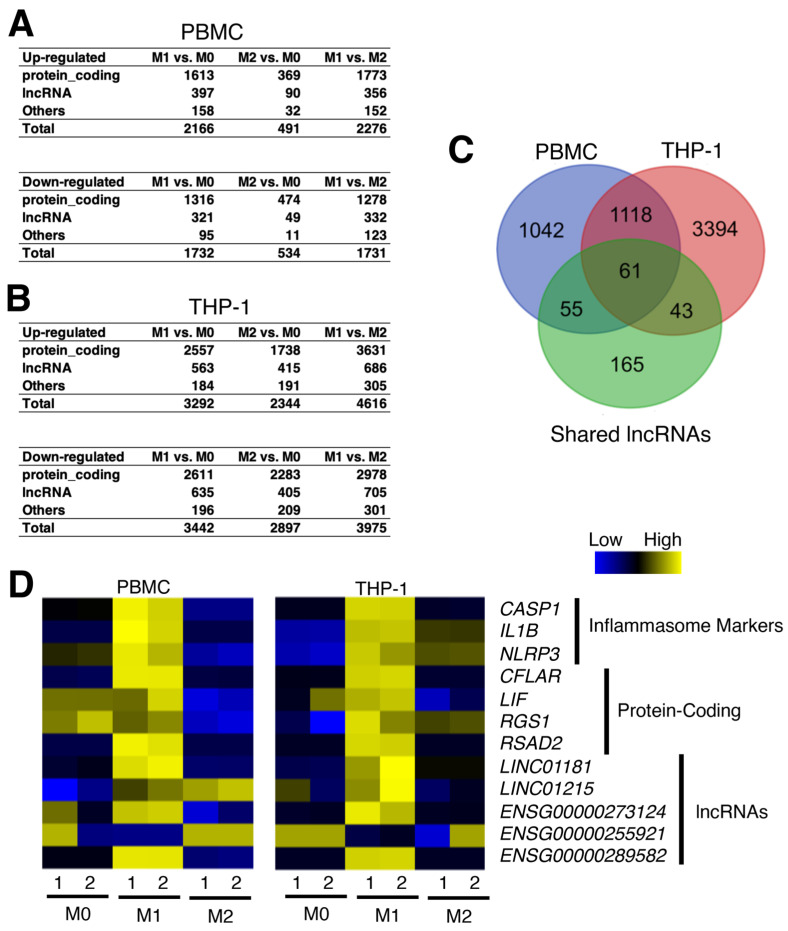
Differentially expressed genes upon polarization of naïve (M0) macrophages to pro-inflammatory M1 and anti-inflammatory M2 macrophages. (**A**,**B**) Numbers of up- and down-regulated genes for protein-coding, lncRNA, and other genes for (**A**) PBMC and (**B**) THP-1 cells (n = 2 per condition). (**C**) Venn diagram comparing M1 vs. M2 macrophages in PBMC and THP-1 cells to the shared up-regulated lncRNAs from the previous subsections. (**D**) Heatmap showing marker and candidate genes from Figure 3B.

**Figure 5 ncrna-09-00050-f005:**
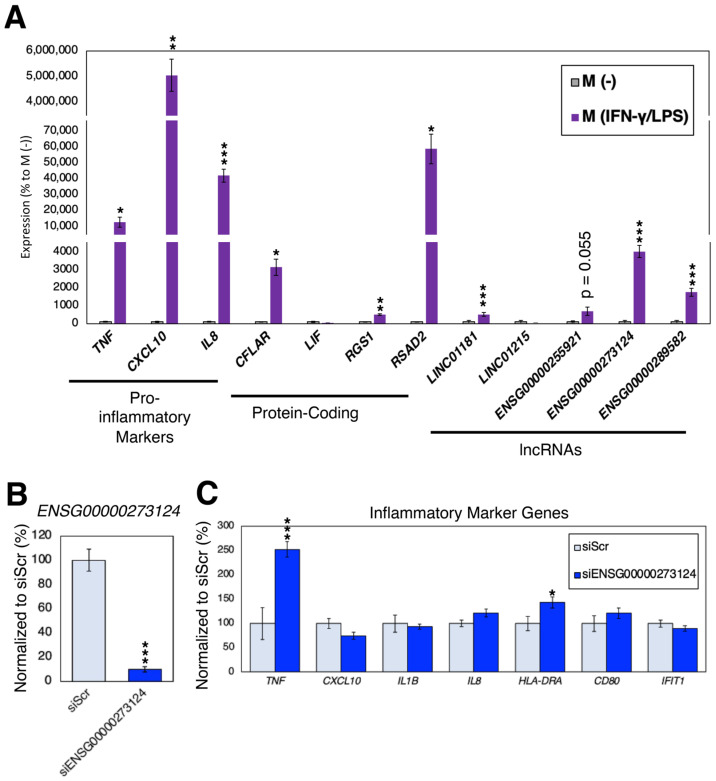
Expression analysis in pro-inflammatory macrophages. (**A**) Expression analysis of pro-inflammatory markers, inflammasome-regulated protein-coding and lncRNA genes. The data were normalized to the expression of M (-) cells. n = 6 biological replicates. * (*p* < 0.05), ** (*p* < 0.01) and *** (*p* < 0.005). (**B**) Silencing of *ENSG00000273124*. The siRNA against random sequence (siScr) was used as control. n = 6 biological replicates. (**C**) Expression analysis of pro-inflammatory markers. n = 6 biological replicates.

**Figure 6 ncrna-09-00050-f006:**
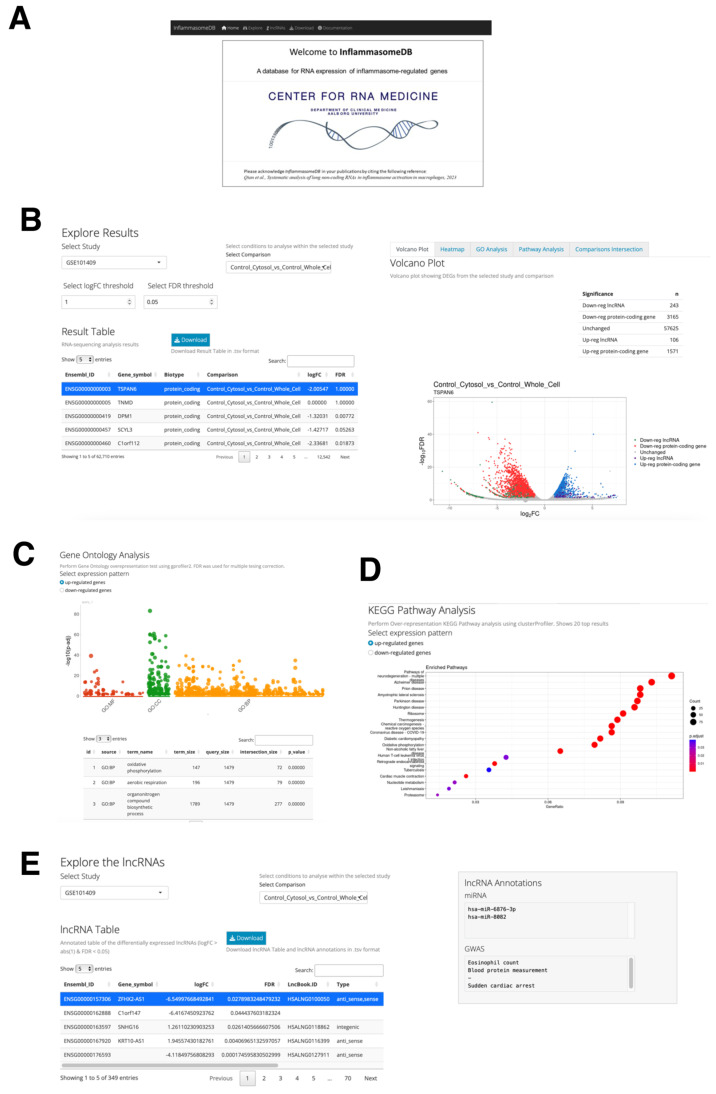
The InflammasomeDB web database. (**A**) Home page of InflammasomeDB. InflammasomeDB can be accessed freely without the need for username and password. (**B**) “Explore Results” page. The user-defined threshold values allow the generation of the Result Table and volcano plot. Both the table and volcano plot are generated based on the user-defined threshold values. (**C**,**D**) Categorization of the DEGs by enriched (**C**) GO terms and (**D**) KEGG pathways for inspecting global changes in signaling pathways. (**E**) lncRNA page. Detailed information about lncRNA genes is provided as a table along with the information obtained from the LncBook 2.0 database. By selecting a lncRNA of interest in the table, the list of possibly bound miRNAs and the information about GWAS are dynamically generated in the gray box in the right-hand side of the window.

## Data Availability

The Appendix A can be found on the GitHub reposi-tory: https://github.com/heartlncrna/Analysis_of_Inflammasome_Study/ (accessed on 5 August 2023). All codes used to generate InflammasomeDB are available on the GitHub repositories: https://github.com/heartlncrna/Analysis_of_Inflammasome_Study (accessed on 5 August 2023); and https://github.com/ccc0321/InflammasomeDB/ (accessed on 5 August 2023).

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
