# Peer review of "Systematic Analysis of Long Non-Coding RNAs in Inflammasome Activation in Monocytes/Macrophages"

_ncrna, 2023, doi:10.3390/ncrna9050050_

Round 1
Reviewer 1 Report
There are a few errors that must be corrected before the manuscript is accepted.
1) In Figure 1C the green ovals are mislabeled. They should be labeled as ORN8L vs. Control (instead of CXCL4 vs Control)
2) Figure 3B shows that only 3 lncRNAs out of the 5 selected lncRNAs were up-regulated with LPS. In lines 158-160 the authors state: “As shown in Figure 3B, all selected commonly up‐regulated protein‐coding and lncRNA genes were up‐regulated in M (LPS) compared to M (‐) cells [...]”. Please correct the text.
3) On lines 228-230 the authors state: “[…] silencing of ENSG00000273124 resulted in up‐regulation of inflammatory marker genes, especially tumor necrosis factor (TNF) (Figure 5C), [...]”. Only two inflammatory marker genes out of 7 selected genes were up-regulated, as shown in Figure 5C. The authors should change the sentence on lines 228-230 to better describe these results.
Author Response
There are a few errors that must be corrected before the manuscript is accepted.
1) In Figure 1C the green ovals are mislabeled. They should be labeled as ORN8L vs. Control (instead of CXCL4 vs Control)
Response: Thank you very much for noticing this mistake. This mistake has been corrected.
2) Figure 3B shows that only 3 lncRNAs out of the 5 selected lncRNAs were up-regulated with LPS. In lines 158-160 the authors state: “As shown in Figure 3B, all selected commonly up‐regulated protein‐coding and lncRNA genes were up‐regulated in M (LPS) compared to M (‐) cells [...]”. Please correct the text.
Response: The above sentence was corrected as follow:
“As shown in Figure 3B, all selected commonly up-regulated protein-coding and three out of five lncRNA genes showed statistically significant up-regulation in M (LPS) compared to M (-) cells as in the case of inflammasome markers (caspase 1 (CASP1), interleukin 1 beta (IL1B), and NLRP3).”
3) On lines 228-230 the authors state: “[…] silencing of ENSG00000273124 resulted in up‐regulation of inflammatory marker genes, especially tumor necrosis factor (TNF) (Figure 5C), [...]”. Only two inflammatory marker genes out of 7 selected genes were up-regulated, as shown in Figure 5C. The authors should change the sentence on lines 228-230 to better describe these results.
Responses: The above sentence was corrected as follow:
“When the markers of inflammation were quantified, silencing of ENSG00000273124 resulted in up-regulation of inflammatory marker genes, tumor necrosis factor (TNF) and major histocompatibility complex, class II, DR alpha (HLA-DRA) (Figure 5C), suggesting that this inflammasome-regulated lncRNA gene might be involved in the polarization of macrophage-like cells to pro-inflammatory macrophage-like cells.”
Reviewer 2 Report
In this manuscript by Qian et al, the authors performed secondary analyses for lncRNA genes on previously generated RNA‐seq data for protein‐coding genes and uncovered distinctive expression profiles of lncRNA genes. They further validated their findings by activating NLRP3 inflammasome in THP‐1 cell line and performed loss‐of‐functional assays of the inflammasome‐regulated lncRNA, ENSG00000273124 and showed its role in polarization of macrophage‐like cells to pro‐inflammatory macrophage‐like cells. Finally, they have built InflammasomeDB web application with expression profiles of protein‐coding and lncRNA genes related to the NLRP3 inflammasome activation. Though the focus of the study was interesting in terms of highlighting the functional roles of lncRNA, the authors may consider few concerns that the authors might consider strengthening their work.
1. The authors reported in the introductions (Lines 31-35) that “Upon activation by various danger signals or cellular stressors, the NLRP3 inflammasome assembles and triggers the conversion of pro‐interleukin (IL)‐1β and pro‐IL‐18 into their mature and biologically active forms, IL‐1β and IL‐18, respectively. The release of IL‐1β and IL‐18 leads to the recruitment and activation of immune cells, such as neutrophils and macrophages, initiating an inflammatory response and pyroptotic cell death”. These lines sounds as if NLRP3 inflammasome assembly and activation is happening elsewhere and consequences of this inflammasome activation is recruiting immune cells and killing them by pyroptotic cells death, which is not true to its context. These lines are misleading and needed much attention. In fact, their predominant presence in the immune cells emphasizes its significant role in immune response.
2. Again, in lines 75-76, is it true that IL-1B activate NLRP3 activation? In fact, NLRP3 inflammasome-mediate IL-1β production can be primed by TLRs activation.
3. What the authors mean when said “Volcano plots compared to the control” in figure 1A. It would be more informative to show the up and down regulated genes of interest in the volcano plot. Moreover, the x-axis and y-axis are unclear in Figure 1A and 2A
4. The info. presented in Fig 6 need clarity for better understanding.
This manuscript may benefit much by use of English language more precisely.
Round 2
Reviewer 2 Report
Comment 3 was partially addressed. Now, given the identities of differentially expressed genes upon treatment with CXCL4, ORN8L, or both, it would be ideal to detail the functional attributes of the highlighted molecules in NLRP3 inflammasome complex. Same with, LPS+MSU. Moreover, it would be worth showing the lnc-RNA ENSG00000273124 in the valcono plots for better identification of the rationale for specifically knocking down this Inc-RNA.
Comment 4 is partially addressed by working on the figure legend. However, it is quite difficult to visualize the figure for better understanding. Though the idea is to provide a overview of the InflammasomeDB, it would worth providing the same with a readable clarity.
Can be improved for better readability
